# Lipid Traffic Analysis reveals the impact of high paternal carbohydrate intake on offsprings' lipid metabolism

Samuel Furse [1,2✉], Adam J. Watkins [3], Nima Hojat [1], James Smith [4], Huw E. L. Williams [5], Davide Chiarugi [6] & Albert Koulman [1,2✉]

In this paper we present an investigation of parental-diet-driven metabolic programming in offspring using a novel computational network analysis tool. The impact of high paternal carbohydrate intake on offsprings' phospholipid and triglyceride metabolism in F1 and F2 generations is described. Detailed lipid profiles were acquired from F1 neonate (3 weeks), F1 adult (16 weeks) and F2 neonate offspring in serum, liver, brain, heart and abdominal adipose tissues by MS and NMR. Using a purpose-built computational tool for analysing both phospholipid and fat metabolism as a network, we characterised the number, type and abundance of lipid variables in and between tissues (Lipid Traffic Analysis), finding a variety of reprogrammings associated with paternal diet. These results are important because they describe the long-term metabolic result of dietary intake by fathers. This analytical approach is important because it offers unparalleled insight into possible mechanisms for alterations in lipid metabolism throughout organisms.

[1] Core Metabolomics and Lipidomics Laboratory, Wellcome Trust-MRL Institute of Metabolic Science, University of Cambridge, Cambridge, UK. [2] Metabolic Disease Unit, Wellcome Trust-MRL Institute of Metabolic Science, University of Cambridge, Cambridge, UK. [3] Division of Child Health, Obstetrics and Gynaecology, Faculty of Medicine, University of Nottingham, Nottingham, UK. [4] School of Food Science & Nutrition, University of Leeds, Leeds, UK. [5] Biodiscovery Institute, University of Nottingham, Nottingham, UK. [6] Bioinformatics and Biostatistics Core, Wellcome Trust-MRL Institute of Metabolic Science, University of Cambridge, Cambridge, UK. ✉email: samuel@samuelfurse.com; ak675@medschl.cam.ac.uk

Beyond the serious risk to their metabolic health, obesity in both men and women has long-term consequences for their offspring through nutritional programming[1–3]. There is increasing evidence showing that the nutritional programming of offspring occurs through changes in lipid metabolism[4] and leads to increased risk of cardio-metabolic disease[5–9]. One contributor to obesity is excess carbohydrate intake. Specifically, high carbohydrate diets have been associated with the emergence of cardio-metabolic disease[10,11] and lower carbohydrate intake with improved recovery[12–14]. One possible explanation is that nutritional programming represents an adaptation to an unbalanced dietary intake in which there is an excess of non-essential nutrients and a deficiency of essential nutrients. However, the effects of a high carbohydrate diet on programming lipid metabolism are not understood. This led us to the hypothesis that a low-protein-high-carbohydrate (LP-HC) diet would alter the programming of lipid metabolism in offspring.

This hypothesis was tested by feeding an isocaloric, non-obesogenic LP-HC diet to the (grand)sires of the experimental groups (mouse model[2,3,15]). This was designed to increase de novo lipogenesis; a high-fat diet would be less useful for testing this hypothesis as it would alter lipid intake as well as biosynthesis. Although the programming effects on de novo lipogenesis were expected to be focused on the offsprings' liver, the products of lipid biosynthesis are typically distributed throughout the organism quickly, especially triglycerides (TG)[16]. Testing this hypothesis, therefore, also required a tool for analysing lipid metabolism and distribution systemically.

However, most computational tools developed to study metabolism are focused on one compartment and thus do not to analyse networks or traffic[17,18]. Analysis of single tissues does not provide a complete picture of systemic programming effects. Furthermore, most of the current tools pivot on substrate-enzyme-product relationships to allow for direct linkage to genes and proteins, rather than the local function of metabolites, making it impossible to characterise a whole system. Equally, lipid metabolism is distinct from amino acid and nucleotide metabolism; lipids are not polymers, vary greatly in structure and comprise components from unconnected sources. We therefore designed a network analysis tool to characterise the number, type and abundance of lipids in and between tissues, referred to as Lipid Traffic Analysis (LTA).

The novel lipid computational tool reported here was used to analyse lipidomics data from liver, serum, brain, heart and adipose tissues for the F1A group (Fig. 1b). Liver, serum, brain and heart samples were used in neonate networks as adipose is too small to be dissectible in these individuals. The connections between the tissues represented the major lipid 'highways' in the organism (Fig. 2a). Lipid Traffic Analysis identified altered lipid metabolism through a switch analysis (which lipids were present and where) and an abundance analysis (quantitative differences between phenotypes). Importantly, these analyses represent the state-of-the-art in the characterisation of lipid metabolism across organs.

We wanted to test the hypothesis that a higher carbohydrate intake in (grand)sires alters lipid metabolism in offspring as this contributes to our understanding of dietary intake and metabolic programming[19,20] and the effects of metabolic disease across generations[4] in a model system. This gives us an insight into possible interventions to improve human metabolic health in familial circumstances.

## Results

A combination of orthogonal techniques, direct infusion mass spectrometry (DI-MS[21,22]) and phosphorus nuclear magnetic resonance ($^{31}$P NMR[23]) was used to profile the lipidome. This approach, known as dual spectroscopy[22], was used to identify and verify the abundance of lipid classes between the two ionisation modes respectively (NMR data for each compartment shown in Supplementary Information) and identified up to 586 lipid variables in positive ionisation mode and up to 564 lipid variables in negative ionisation mode in liver, brain, heart and adipose homogenates and in serum.

**Lipid Traffic Analysis: a novel computational tool for the network analysis of lipid metabolism.** The first stage in developing a computational tool to analyse lipid metabolism systemically was to categorise lipid variables according to where they were found. The relationship between adjacent lipid

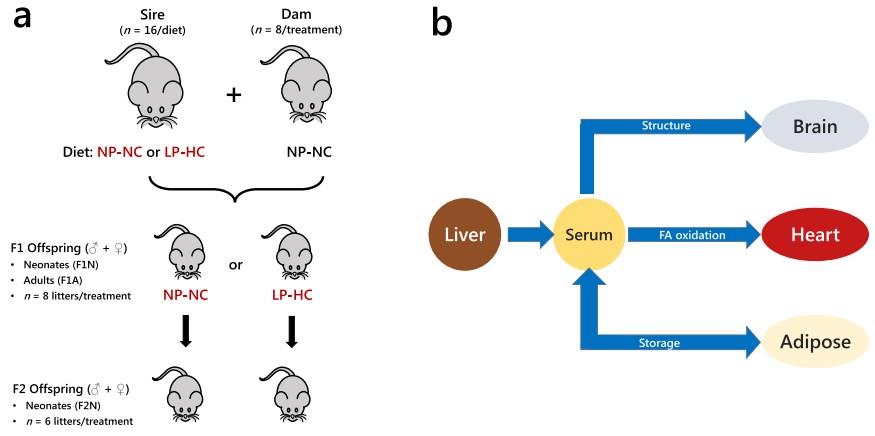

**Fig. 1 The mouse model and tissues used for lipid traffic analysis associated with de novo lipogenesis. a** Schematic representation of the mouse model showing the generation of programmed offspring across two generations. **b** The network that describes the lipid traffic associated with de novo lipogenesis from the liver to termini (CNS, heart and adipose) via the serum. The termini represent traffic for structural purposes (CNS), fatty acid oxidation (heart) and storage (adipose). This metabolic relationship between tissues was used as the structure of the network for all analyses in the present study. Adipose was not available for neonates and thus networks for F1N and F2N do not include this tissue. Cerebellum and right hemispheres of the brains of the F1A and F2N groups, enabling separate analysis of cortices and the cerebellum in these groups. NP-NC refers to a diet of normal protein-normal carbohydrate where LP-HC refers to a low protein-high carbohydrate diet. The NP-NC and LP-HC are the same as NN and LL groups used in earlier studies[2,15].

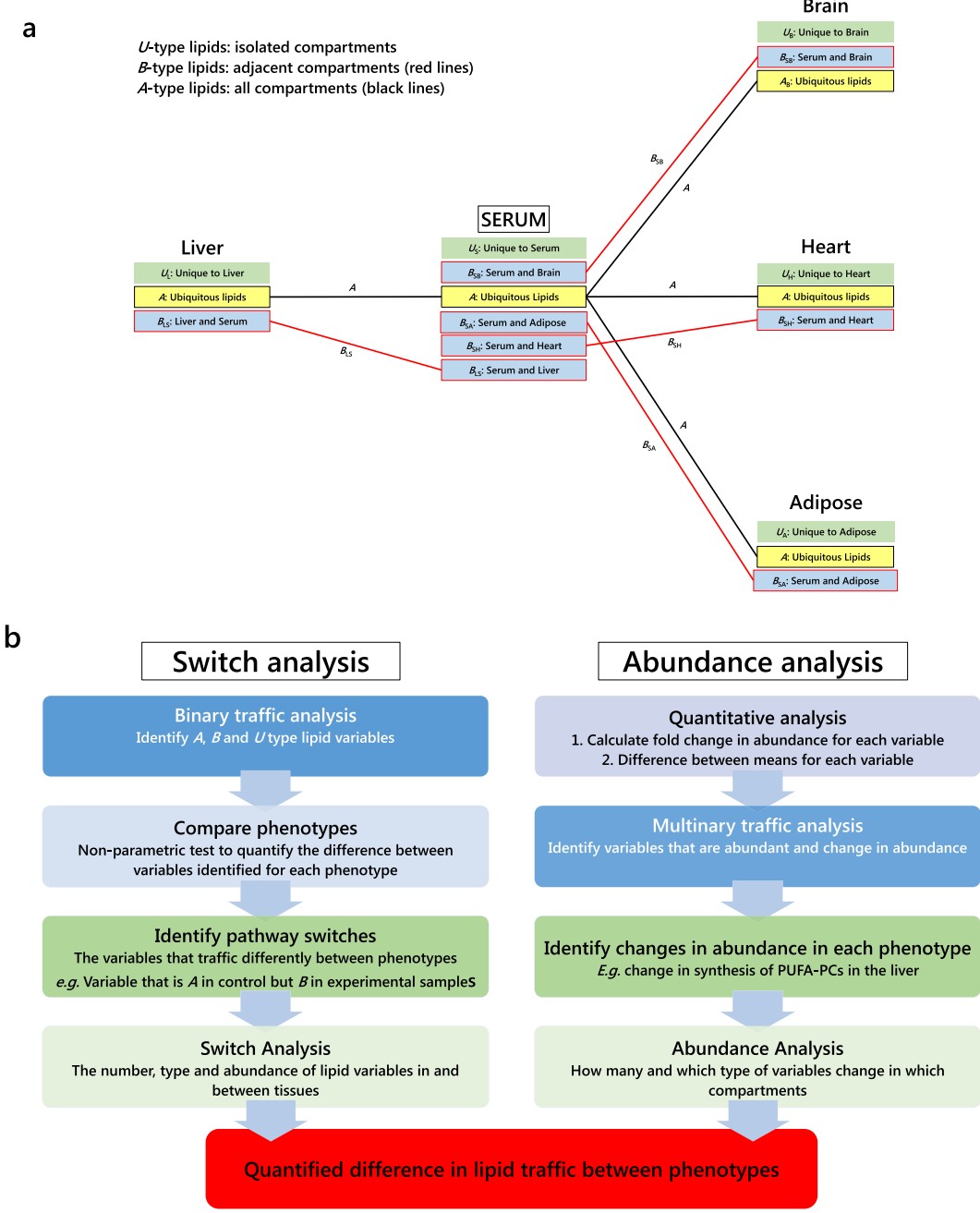

**Fig. 2 Structure of traffic analysis for quantifying changes in lipid metabolism. a** Categorisation of lipids according to where they are found; **b** flow chart of traffic analysis showing the gross structure of the analysis. **A**, **B** and **U** are categories representing variables that appear in all compartments, in pairs of adjacent compartments and in only one compartment, respectively. Subscripts to these categories are pairs of one-letter codes indicating the direction of the traffic (reading left to right). Red connections show B-type lipid connections. Black connections show A-type lipid connections. The two strands of the flow chart represent separate analyses that use the same R code (see SI). Equations for the quantitative analysis are shown in Eqs (1) and (2).

compartments is key in the biological network (Fig. 1). Some lipid variables were found in *all* compartments, others in two adjacent compartments and others in one compartment only (Fig. 2a). These we refer to as **A**, **B** and **U** type lipids (or categories), respectively. Novel code written in R for identifying such lipids is described in 'Methods' and can be found in the Supplementary Information. The basis of these categories was that they represented the intersections between lipid compartments, i.e. stations in the network. Distinct patterns of the presence of lipid species that appear in adjacent compartments or ubiquitously can represent systemic responses. Different axes between compartments (e.g. liver-serum, serum-heart) can be considered and

physiological metabolic functions compared. This relationship was characterised in the present study using unlabelled species as an average over longer periods, e.g. stage of development. These categorisations of the lipids were then used to construct a Lipid Traffic Analysis from two different perspectives, namely a quantitative abundance analysis and a binary switch analysis (Fig. 2b). Both of these represent novel analyses presented in this work.

**Switch Analysis.** The Switch Analysis developed in the present study identified the lipids that are above or below the limit of detection in a given compartment (**U**), between adjacent

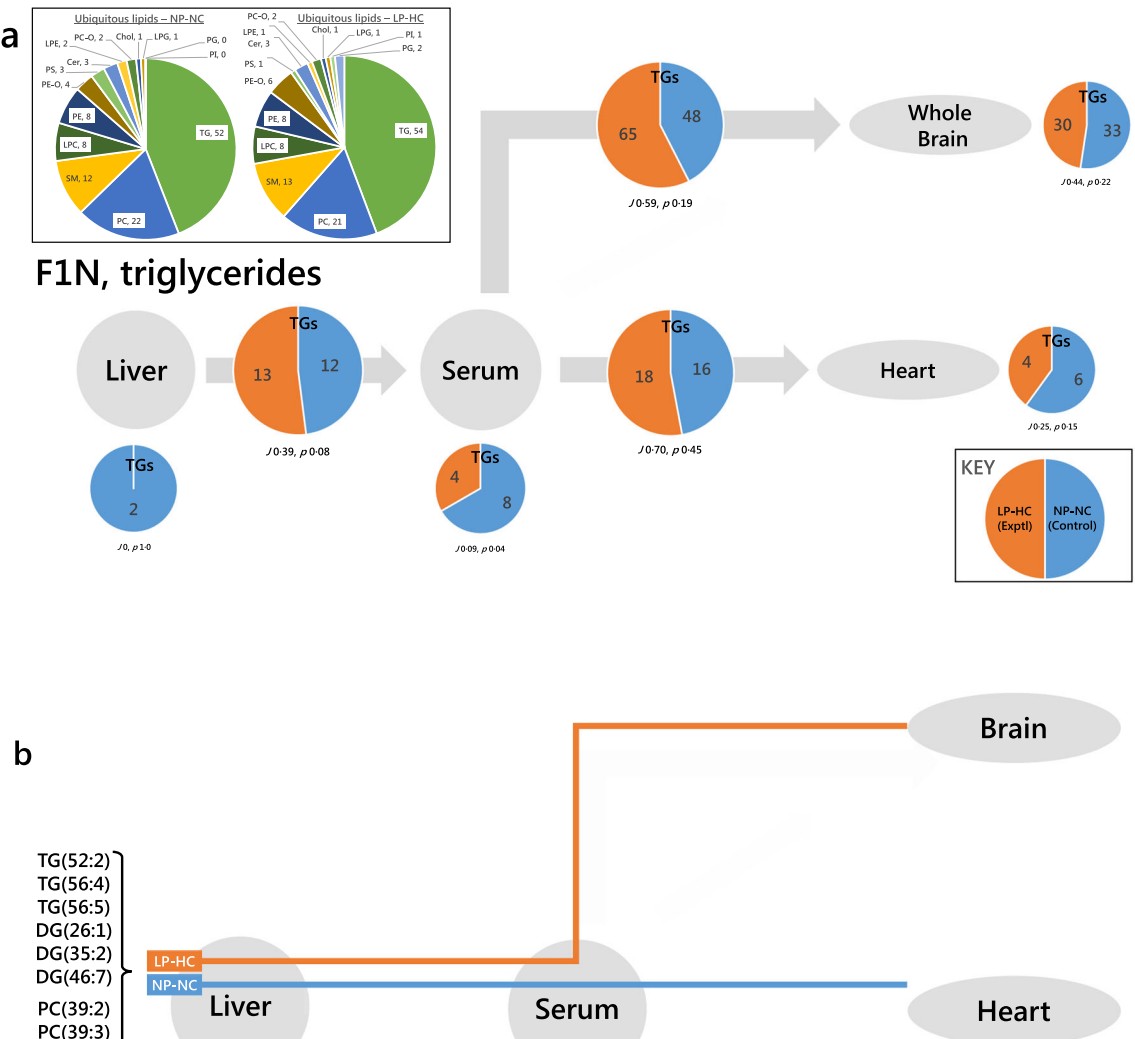

**Fig. 3 Switch analyses of triglyceride variables in the neonatal F1 offspring (F1N) of fathers fed a normal (NP-NC) or a low-protein, high carbohydrate (LP-HC) diet, measured by mass spectrometry. a** Traffic analysis of triglyceride variables; **b**, routing diagram of the switch analysis of triglyceride (measured in positive ionisation mode) and phosphatidylcholine (measured in negative ionisation mode) variables in F1 neonatal mice re-routed from the serum-heart in the control group to the serum-brain in the experimental group. TG and PC variables on the serum-heart axis in the control (NP-NC, blue) group of F1Ns that are found on the serum-brain axis of the experimental (LP-HC, orange) group of F1Ns but not their serum-heart axis. The pie charts in the insert show the number of ubiquitous lipid variables for that network, for each phenotype. Larger pie charts represent lipids found in two adjacent compartments (**B**-type lipids). Smaller pie charts represent lipids found in isolated in compartments (**U**-type lipids). *J* represents the Jaccard-Tanimoto coefficient for the comparison, with accompanying *p* value, as a measure of the similarity between the variables identified in the two phenotypes for each comparison. The *p* value shown represents the probability that the difference between the lists of variables for the two phenotypes occurred by random chance. Orange is used for the LP-HC group, whereas blue is used for the NP-NC group. Cer, ceramide; Chol, cholesterol; DG, diglyceride (water-loss product from fragmentation in source); LPC, *lyso*-phosphatidylcholine; LPE, *lyso*-phosphatidylethanolamine; LPG *lyso*-phosphatidylglycerol; PA, phosphatidic acid; PC, phosphatidylcholine; PC-O, phosphatidylcholine plasmalogen; PE, phosphatidylethanolamine; PE-O, phosphatidylethanolamine plasmalogen; PG, phosphatidylglycerol; PI, phosphatidylinositol; PS, phosphatidylserine; SM, sphingomyelin; TG, triglyceride.

compartments (**B** type lipids, e.g. liver-serum axis) or ubiquitously (**A** type lipids). This, therefore, represented lipids that were switched on or off with respect to a measurement threshold. The Switch Analysis requires only straightforward lipidomics data rendered as zero or non-zero values, e.g. relative abundance. The Switch Analysis of TGs in the F1N group is shown in Fig. 3a. Pie chart segments show the number of TG variables in each phenotype. Jaccard-Tanimoto coefficients (JTC, with accompanying *p* values) were used to characterise the similarity between the compared groups. The JTC indicated the proportion of variables that appeared in the two groups[24,25] whereas the *p* value indicated what differentiated them; a *p* < 0.5 indicated there were variables

unique to both groups, whereas *p* > 0.5 meant that only one group could have any unique variables. Thus, a JTC of 0.67 with a *p* of 1 indicated that two thirds of variables appear in both groups, with the other third of the variables only appearing in one group.

We first tested the hypothesis that a high carbohydrate diet shaped lipid metabolism in offspring by investigating TGs, as several of these are well-established markers of de novo lipogenesis (DNL) and thus are affected by changes in lipogenesis (Fig. 3a. Major TGs on the liver-serum axis were also found on the serum-brain and serum-heart axes in F1N NP-NC (control) mice. This led us to ask whether any of those variables were routed differently according to phenotype.

Specifically, it was observed that the number of TG variables that appeared on the serum-brain axis in F1N of the LP-HC group was both more and different to than the NP-NC group (48:65, *J* 0.59, *p* 0.19, Fig. 3a). We therefore tested the hypothesis that variables found in the control group serum-heart axis were also found in the serum-brain axis of the high carbohydrate but not control group. We plotted the presence of the appropriate variables in the Switch Analysis as a network connections diagram similar to those used in wiring diagrams or on the London Underground map (Fig. 3b). This type of plot shows the routing of the variables and is thus referred to here as a routing diagram. This analysis showed that six variables were found to be on the serum-heart axis of the control group but not on the serum-brain axis of the experimental group (LP-HC). These results implying that those variables were re-routed in this phenotype (Fig. 3b).

In the F1 Adults, whose network included adipose tissue, we found that all of the TG variables that appeared on the liver-serum axis in F1A NP-NC also appeared on one or more of the serum-cerebellum, serum-right brain, serum-heart or serum-adipose axes for the NP-NC group (Supplementary Table S1). This suggests that there may be control over how lipid distribution is gated through the system. The larger number of variables found on the serum-adipose axis in F1A adult NP-NC mice compared to LP-HC, and the larger number of TG variables found on the serum-cerebellum and serum-right brain axes suggested to us that TGs were re-routed from the adipose to the CNS in F1 adults due to the low protein, high carbohydrate paternal diet (29:36, *J* 0.8, *p* 0.36, Fig. 4a). This was true for at least two variables, DG(33:1) and TG(52:5) (Table S2, columns 3–5). However, this left several variables unaccounted for. Furthermore, there was a difference in the number of variables in the NP-NC and LP-HC groups on the liver-serum axis (Fig. 4a). We therefore tested the hypothesis that these seven variables were associated with different routing in the two phenotypes. The seven variables that distinguished NP-NC from LP-HC on the liver-serum axis of F1A (shown in pale green cells in Table S2) were all either found in the serum-cerebellum/ right brain or serum-heart axes of the LP-HC phenotype, but not the NP-NC phenotype. This showed that an infrastructure and set of controls that govern the routing of lipids between tissues (gating) exists in these systems.

Trafficking of TGs was also investigated in F2N individuals (Fig. 4b). This analysis showed that there were a considerable number of variables unique to the cerebellum in the NP-NC group and not present in the LP-HC group. The network analysis used in this study revealed that these variables were not found elsewhere in the system. This result is remarkable because the cerebellum does not typically use FAs for energy, raising questions about why TGs are in the cerebellum and why they are only present in one phenotype. 14 of the 20 TGs identified comprise FAs with an odd number of carbons in the chain (Table S3). This result forms part of a wider characterisation of the relationship between fats and the CNS, with evidence for re-routing of TGs to the CNS from the heart in the LP-HC phenotype (Figs. 3 and 4). A close or complicated relationship of the CNS with energy supply by TGs is counter-intuitive because the principal carbon source of the CNS is glucose and not fat, and even under starvation conditions, only a small proportion of the ATP used in the CNS is made from energy released from primary metabolism of fats.

As the traffic of TGs differed between phenotypes, the hypothesis that the traffic of lipids associated with cell structure, such as phosphatidylcholine (PC), was associated with a LP-HC diet in (grand)sires across F1N, F1A and F2N groups was also tested. The results of Lipid Traffic Analysis in F1N indicated the possibility of a re-routing of PC variables according to phenotype (Supplementary Fig. S1A). Specifically, more PC

variables were found on the serum-brain axis in the LP-HC group, where the opposite was the case for the serum-heart axis. There were at least four variables that distinguished the NP-NC serum-heart axis and the LP-HC axis, all of which also distinguished the serum-brain axis in the LP-HC group from the NP-NC (Fig. 3b).

In the F1A network, eight PCs were found on the serum-heart axis in the NP-NC group that were not found on the same axis of the LP-HC group (Fig. 5a, 56:51, *J* 0.83, *p* 0.41). A routing diagram showed that two of these were re-routed to the CNS in the LP-HC (e.g. PC(43:2), Fig. 5b). This analysis suggests that several effects differentiate the two groups through the system, including biosynthesis and transport. It also showed that the sections of the CNS tested did not show the same shifts in distribution according to phenotype. This showed that distribution within the CNS is also altered as well as in the periphery. For example, we found that PC(33:4) was only found in the LP-HC phenotype, whereas PC(42:4) was found throughout the NP-NC phenotype but only in the LP-HC liver. This, with changes to PE that are similar (e.g. PE(41:1), Fig. 5b), suggested that several modulations to PC and PE metabolism associated with this phenotype. It is also consistent with long-standing evidence that PC and PE are used as a means for storing/transporting polyunsaturated FAs such as arachidonic acid[26].

PC traffic in the F2N network (Supplementary Fig. S1B) was characterised by more PC variables being produced in the control (NP-NC) group (Table S4). Seven additional variables were found on the Liver-Serum axis of the F2N NP-NC group. Several of these, PC(39:3, 40:4, 40:6, 41:4), were also found on the serum-cerebellum and serum-right brain axes (Supplementary Table S4), suggesting that the grandsires' dietary balance influenced the biosynthesis and distribution of phosphatidylcholine in F2 offspring. Full Switch analysis is shown in Supplementary Data 1.

**Abundance Analysis.** Perhaps the most striking result from the Switch analysis (vide supra) is the association of the grandsire's diet to the biosynthesis and gating of lipid variables from the serum-heart axis to the serum-CNS axes. This was observed particularly clearly with phosphatidylcholines (Figs. 3 and 5) for F1N and F1A, with a distinct difference in the number of PC variables trafficked to both the heart and CNS in F2N mice (Supplementary Fig. S1). This led us to the hypothesis that there would be a change in the abundance of the ubiquitous PCs associated with the phenotype, i.e. the PCs found in all tissues would have a difference in abundance between phenotypes. We designed an Abundance Analysis to be able to test this.

Two numerical dimensions were used in the abundance analysis. One of these describes the margin of the difference in abundance between the two phenotypes (Eq. 1) and the other the magnitude of the difference (error normalised fold change, Eq. 2). In these equations $\bar{x}_C$ is the mean of values for that variable in the control group (NP-NC), $\bar{x}_E$ is the mean of values for the experimental group (LP-HC), '*a*' is the standard deviation of the values of the abundance of the given variable in the NP-NC group and '*b*' is the standard deviation of the values of the same variable for the LP-HC group.

$$\text{Margin change} = \bar{x}_E - \bar{x}_C \tag{1}$$

$$\text{Error normalised fold change} = \frac{\log_{10}\left(\frac{\bar{x}_E}{\bar{x}_C}\right)}{\left(\sqrt{\frac{(a^2+b^2)}{2}}\right)} \tag{2}$$

The margin change is interpreted with a *p* value calculated using a Student's *t*-test (see 'Methods'), whereas the fold change

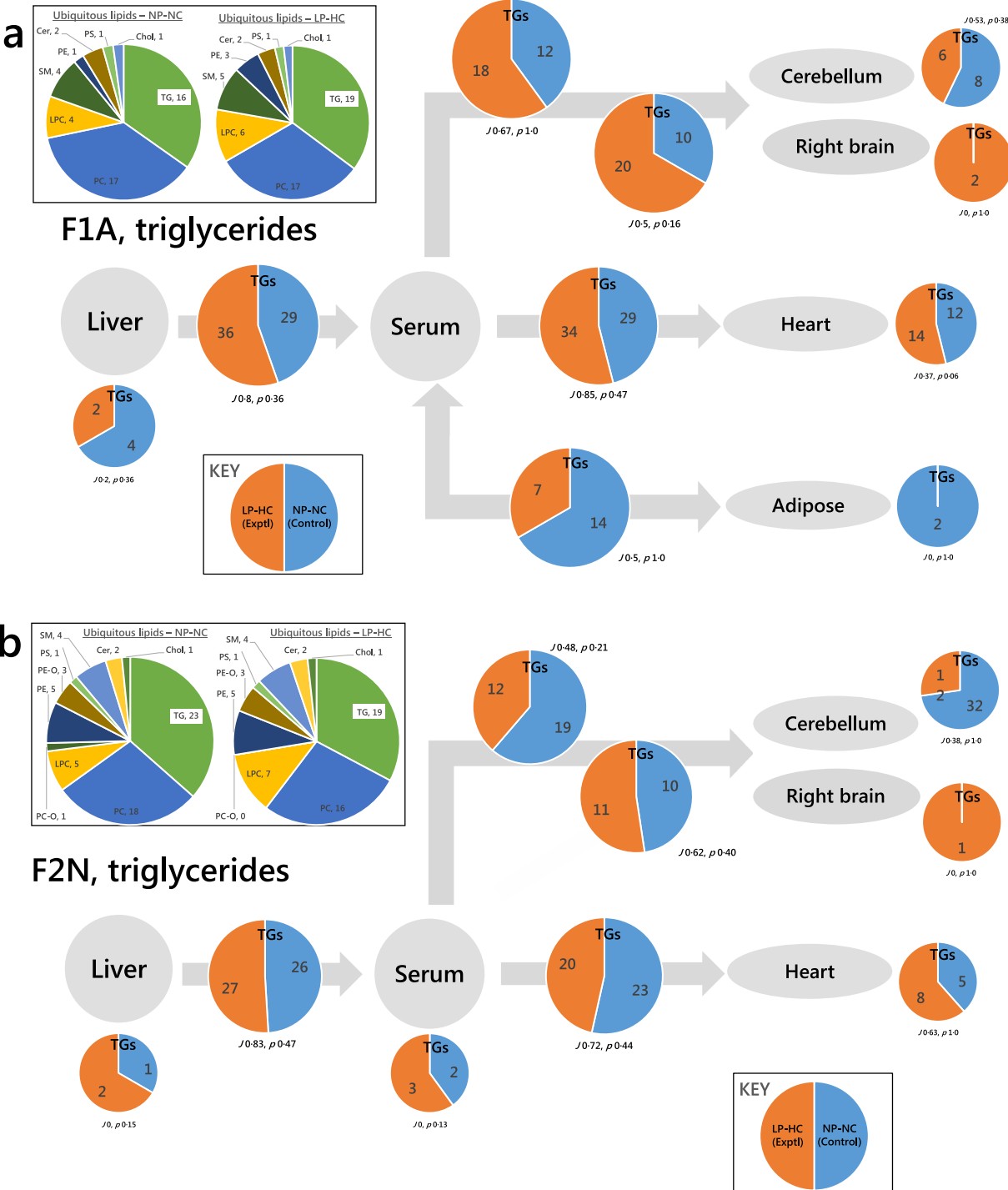

**Fig. 4 Switch Analyses of triglyceride variables in the adult F1 (F1A) and neonatal F2 (F2N) offspring of fathers fed a normal (NP-NC) or a low-protein, high carbohydrate (LP-HC) diet, measured by mass spectrometry in positive ionisation mode. a** F1 Adults (F1A); **b** F2 Neonates (F2N). The pie charts in the insert show the number of ubiquitous lipid variables for that network, for each phenotype. Larger pie charts represent triglyceride variables found in two adjacent compartments (**B**-type lipids). Smaller pie charts represent triglyceride variables found in isolated in compartments (**U**-type lipids). *J* represents the Jaccard-Tanimoto coefficient for the comparison, with accompanying *p* value, as a measure of the similarity between the variables identified in the two phenotypes for each comparison. The *p* value shown represents the probability that the difference between the lists of variables for the two phenotypes occurred by random chance. Orange is used for the LP-HC group whereas blue is used for the NP-NC group. Cer, ceramide; Chol, cholesterol; DG, diglyceride (water-loss product from fragmentation in source); LPC, *lyso*-phosphatidylcholine; LPE, *lyso*-phosphatidylethanolamine; LPG *lyso*-phosphatidylglycerol; PA, phosphatidic acid; PC, phosphatidylcholine; PC-O, phosphatidylcholine plasmalogen; PE, phosphatidylethanolamine; PE-O, phosphatidylethanolamine plasmalogen; PG, phosphatidylglycerol; PI, phosphatidylinositol; PS, phosphatidylserine; SM, sphingomyelin; TG, triglyceride.

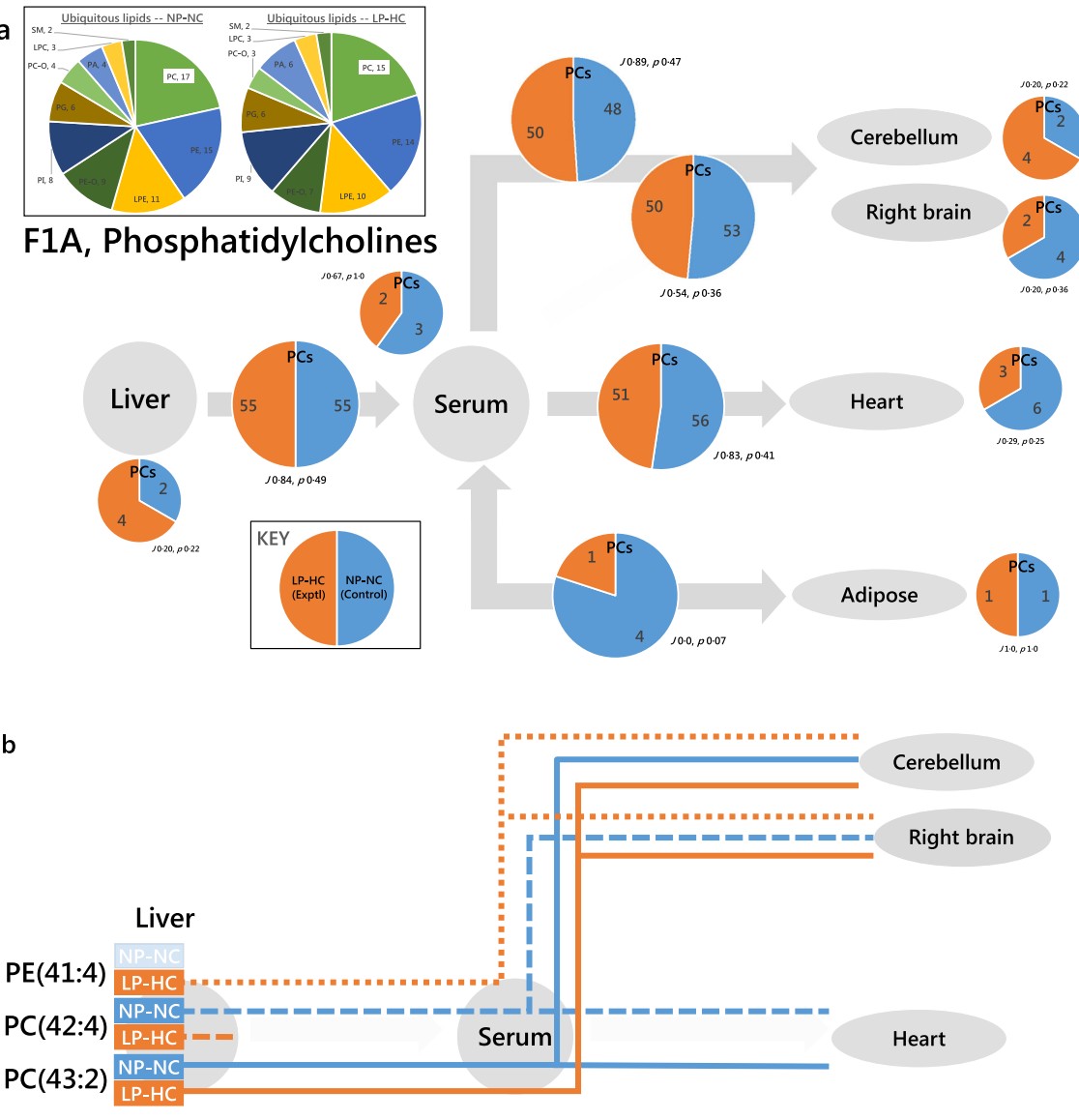

**Fig. 5 Switch Analyses of phospholipid variables in the adult F1 (F1A) offspring of sires fed a normal (NP-NC) or a low-protein, high carbohydrate (LP-HC) diet, measured by mass spectrometry in negative ionisation mode. a** All phosphatidylcholines; **b** network diagram showing the distribution of PE (41:4) (dotted line), PC(42:4) (dashed line), PC(43:2) (solid line). The pie charts in the insert show the number of ubiquitous lipid variables for, for each phenotype. Larger pie charts represent PC variables found in two adjacent compartments. Smaller pie charts represent PC variables found in isolated in compartments. $J$ represents the Jaccard-Tanimoto coefficient for the comparison, with accompanying $p$ value, as a measure of the similarity between the variables identified in the two phenotypes for each comparison. The $p$ value shown represents the probability that the difference between the lists of variables for the two phenotypes occurred by random chance. Orange is used for the LP-HC group whereas blue is used for the NP-NC group. LPC, *lyso*-phosphatidylcholine; LPE, *lyso*-phosphatidylethanolamine; PA, phosphatidic acid; PC, phosphatidylcholine; PC-O, phosphatidylcholine plasmalogen; PE, phosphatidylethanolamine; PE-O, phosphatidylethanolamine plasmalogen; PG, phosphatidylglycerol; PI, phosphatidylinositol; PS, phosphatidylserine; SM, sphingomyelin; TG, triglyceride.

has a built-in confidence interval through a calculation of the propagated error. The margin changes and accompanying probabilities were used to identify the variables that describe the difference in lipid metabolism between phenotypes. The error normalised fold change (ENFC) was used to quantify this.

The Abundance Analysis found that PE(40:2) and (40:3) were more abundant in the livers of the LP-HC group of F1N mice ($p = 0.0005$ and $0.001$, respectively). This was reflected in the

abundance pattern in F2Ns, but not F1As (ENFC, plotted in Fig. 6a). PE(34:1) was less abundant in the livers of LP-HC F2Ns ($p = 0.0021$) and PE(36:3) less abundant in the serum of LP-HC F2Ns ($p = 0.0006$), however, they were in general more abundant in the CNS of LP-HC F2Ns (ENFC plotted in Fig. 6b). Two commonplace PC isoforms (38:1 and 38:4) were both less abundant in the CNS of LP-HC F2Ns ($p = 0.0009$ (Cerebellum) and $0.0015$ (right brain) respectively), with mixed effects noted

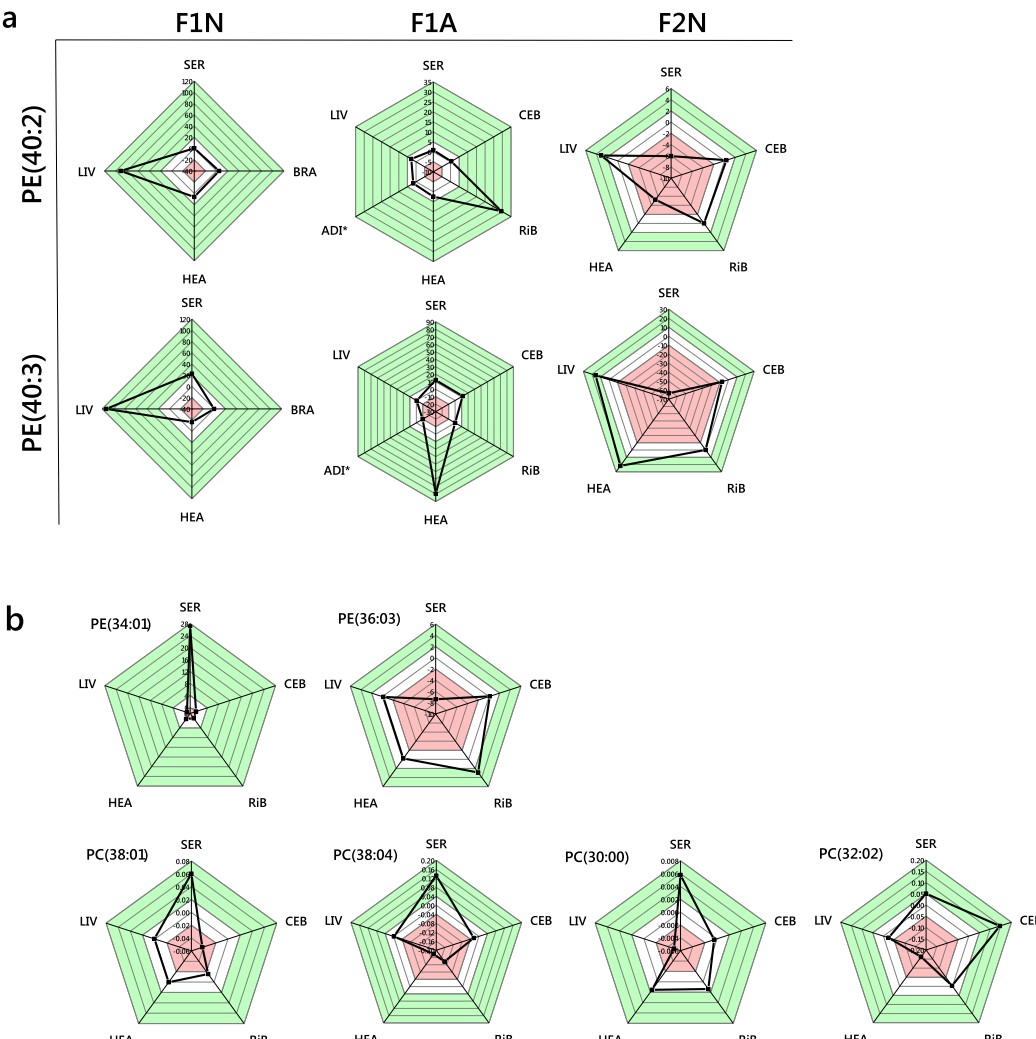

**Fig. 6 Radar plots of the Error Normalised Fold Change in abundance of phosphatidylcholine and phosphatidylethanolamine lipid variables associated with a high carbohydrate dietary intake of (grand)sires. a** PE(40:2, 40:3) were identified as more abundant in F1N livers using statistical approaches, this was followed through all generational groups; **b** PC and PE variables whose abundance in the CNS changes in a manner associated with the dietary phenotype. The white areas represent the 0 point and one division above and below this. The red areas represent values more negative, and the green areas values more positive than this. ADI, adipose; BRA, brain; CEB, cerebellum; HEA, heart; LIV, liver; RiB, right brain; SER, serum. PC, phosphatidylcholine; PE, phosphatidylethanolamine. *Sample treated with petroleum spirit to concentrate phospholipid fraction (see 'Methods').

for PC(30:0 and 32:2) in the same tissues ($p = 0.001$ and 0.002 respectively, ENFC plotted in Fig. 6b). Importantly, 38:1 and 38:4 are commonplace and typical isoforms of PC found in the CNS, as are the PEs. When taken with the Switch Analysis results (Figs. 3 and 5), this suggests that commonplace isoforms of PC are rerouted away from the CNS and replaced by more recondite ones, e.g. PC(39:2, 43:2), and PE (e.g. 36:3), in the grand-offspring of fathers fed a high carbohydrate diet. The higher abundance of PE(34:1) in the circulation of F2N (ENFC = 27.2, Fig. 6b) and PE (40:2, 40:3) in the liver of F2N (ENFC = 3.1, 15.8) without a consummate increase in the CNS suggests that despite the difference between phenotypes, PEs(34:1, 40:2, 40:3) are handled differently to PE(36:3) in LP-HC offspring.

It is also clear from the Switch Analysis that TGs are trafficked differently in these two systems (Figs. 3 and 4), including evidence for TG variables being rerouted (Fig. 3). This led us to test the hypothesis that a higher carbohydrate diet consumed by fathers altered de novo lipogenesis (DNL) in offspring. We elected to use a targeted approach for testing this, using known markers of DNL[16] and reference variables not associated with

DNL. The abundance of all DNL TGs was typically much higher in CNS tissue in the LP-HC group (Fig. S2A–C). This was especially clear in F1A individuals (Fig. S2B), where all of the DNL variables were much more abundant in the Right Brain of LP-HC mice. The abundance of a dietary TG and a species made endogenously (TG(54:4) and cholesterol, respectively) were also higher in F1A the CNS. However, reference species not associated with DNL, such as phospholipids were not more abundant in the LP-HC phenotype (Fig. S2B), suggesting that the change in lipid traffic is not restricted to DNL species only. These data are consistent with the Switch Analysis (Figs. 3 and 4).

## Discussion

This study was motivated by the hypothesis that a higher carbohydrate, lower protein intake in the paternal grandsire diet influences the lipid metabolism of their offspring. Programmed changes to DNL were expected. Detailed molecular lipid surveys of several tissues associated with these two phenotypes (NP-NC, control; LP-HC, experimental) were analysed using a novel

computational/bioinformatics tool for analysing lipid metabolism as a network (Lipid Traffic Analysis). This showed that the number, type and abundance of lipid variables in and between tissues differed between phenotypes and generations. The diets were designed to test how DNL and thus metabolic activity in the liver could be programmed. A focused characterisation of the lipid metabolism across two succeeding generations from sires fed in this way revealed that both triglyceride (TG) and phosphatidylcholine (PC) metabolism were altered throughout the network by this nutritional programming, and over two generations. However, the changes to lipid traffic and biosynthesis do not appear to be restricted to species associated with DNL, suggesting the effects are wider than those in the hypothesis.

In particular, the evidence for both TG and phospholipid variables being re-routed to the CNS from the heart and adipose is striking. This was observed in both F1N individuals (Fig. 3b) and F1As (Fig. 4a). This showed that both the structural molecules and molecules that supply of energy (TGs) are associated with LP-HC programming.

The change in traffic of TGs offers evidence for changes to metabolic control as a result of TGs crossing the blood-brain barrier. The phenomenon of metabolic effects associated with TGs in the CNS has been observed through central leptin and insulin receptor resistance[27], however, the variety of TGs has not previously been described. The present study shows that TGs associated with dietary intake are routed to the CNS in F1Ns whose fathers ate a high carbohydrate diet (LP-HC), whereas they are routed to the heart in a normal diet (TG(52:2, 56:4, 56:5), Fig. 3b). This result is insightful because it shows that it is not simply dietary intake that shapes how metabolism is organised. These data suggest that intake shapes how lipid distribution is organised. This in turn hints at changes in infrastructure that offer a mechanism for possible changes in fuel uptake at the two termini (brain, heart) between phenotypes.

The same analysis in the F2N group showed that there were 14 odd-chain-containing TGs in the CNS of the NP-NC but not LP-HC group (Fig. 4b, Table S3), showing that changes to lipid distribution are evident at least two generations hence. This is strong evidence that the apparent controls over lipid distribution are associated with parental and even grand-parental diet with local effects some way away from the liver.

Specifically, this evidence suggests that a high carbohydrate diet in fathers may programme their infants for insulin and leptin resistance. Evidence to link changes to lipid processing in the brain with metabolic disease (review[28]) suggests that there may be several possible specific effects. Thus, the present study shows that the nutritional programming associated with a non-obese phenotype tends towards metabolic disease. Furthermore, this is generally consistent with recent work showing that adipose volume and morphology in non-obese individuals is associated with metabolic disease outside of obesity[29,30] (reviews[31,32]). This suggests that unbalanced paternal nutrition as well as excess nutrition can result in altered metabolism over two generations. This is revealing because it offers a possible mechanism for metabolic disorders in individuals with a healthy body mass index.

As well as changes to energy storage and distribution, there is evidence for structural change in these systems too. There are several changes in the traffic of phospholipids that differ from those of triglyceride variables. We have found evidence for changes to the traffic of both PC and PE (Fig. 5). Changes to PC traffic centre on the CNS and heart, with some striking differences in the gating of some phospholipid variables between phenotypes. The examples shown in Fig. 5b show that the gating of polyunsaturated and odd-chain-containing phospholipids is altered between phenotypes. We suggest that this evidence shows

that several factors differ between phenotypes, many of which are worthy of further investigation. First, biosynthesis in the liver is affected. This changes the availability of these lipids for trafficking to other parts of the system. Second, some lipids are produced in the liver but do not appear elsewhere, suggesting that transport of lipids from the liver into the circulation is altered between phenotypes. Third, the termini in which they are found (heart/CNS) differs between feeding groups. This suggests uptake in tissues differs between the two groups. Between them, these changes show that a number of aspects of lipid metabolism and distribution were altered as a result of paternal dietary intake. Further studies in this area could include an analysis of gene expression of proteins involved in these processes, such as transporters in the blood-brain barrier that are known to transport polyunsaturated-fatty-acid-containing PCs into the CNS[33,34]. This could also be used gain insight into the expression of lipases that shuffle fatty acids between phospholipids[35,36] and into proteins that remove single carbons from fatty acids to give odd-chain-containing species, such as PC(43:2), e.g. *Hacl1*[37–39].

The evidence of shifts in lipid metabolism in the abundance analysis (Fig. 6) from the present study suggest that the chow diet fed to F1 offspring softens some of the effects of the low protein, high carbohydrate diet for the F2 generation. This suggests that there is scope for a type of reprogramming. Specifically, the pronounced increase in the abundance of PE(40:2, 40:3) in the livers of F1N LP-HC mice is lower in the F2N LP-HC group. Similar patterns are observed for the ENFC of SM(36:1) and TG (48:0) between F1N and F2Ns (Fig. S2). It is not clear from the present study precisely what causes this, however, with appropriate experimental design, hypotheses for programming and reprogramming could be tested. Anabolic hormones such as insulin regulate the release and reuptake of lipids from one organ to another[4,40,41], making Traffic Analysis a powerful tool in characterising the change in lipid metabolism and accumulation. Currently, these analyses are limited to comparisons of blood plasma or serum samples. The new method described here is therefore capable of uncovering new biological meaning in lipid metabolism, and relating lipids in different compartments in a way not possible for simple comparisons.

In conclusion, this study has shown that the hypothesis that lipid metabolism is altered in offspring as a result of unbalanced dietary intake by grandsires is correct, however in a wider manner than expected. The biosynthesis of both TGs and PCs is altered in the liver, with a particular increase in TG traffic reaching the CNS. Furthermore, it is associated with all TGs and not exclusively those associated with DNL. This work shows that a non-obesogenic high carbohydrate, low-protein diet consumed by fathers influences lipid metabolism in offspring over at least two generations. Specifically, the distribution of both triglycerides and phosphatidylcholines is altered in F1 and F2 generations.

The change in supply of phospholipids is consistent with changes in the physical behaviour in membranes of the cardiovascular system that are associated with cardio-metabolic disease. A change in the supply of TGs may also be consistent with pronounced molecular changes in the shift from fatty acid oxidation to glucose metabolism associated with cardiac hypertrophy. These results are, therefore, important because they suggest a molecular mechanism for the emergence of cardio-metabolic disease.

The network approach to the analysis of lipid metabolism reported here was essential for identifying changes in lipid metabolism that occur across pathways (TG/PL) and with components from different sources (endogenous/dietary), however, further work using transcriptomics techniques is required to understand changes to the infrastructure associated with lipid metabolism. These and other techniques can be used to show how

the damaging changes to lipid metabolism that have been identified can be reversed.

## Methods

**Materials, animals, consumables and chemicals.** Purified lipids were purchased from Avanti Polar lipids Inc. (Alabaster, Alabama, US). Solvents and fine chemicals were purchased from SigmaAldrich (Gillingham, Dorset, UK) and not purified further. Mice were purchased from Harlan Laboratories Ltd (Alconbury, Cambridgeshire, UK). Hormones were purchased from Intervet (Milton Keynes, UK).

**Animal model.** All procedures were conducted in accordance with the UK Home Office Animal (Scientific Procedures) Act 1986 and local ethics committees at Aston University. Animals were maintained at Aston University's biomedical research facility as described previously[2] and is shown in Fig. 1a in the context of the present study. Briefly, entire and vasectomised 8-week old C57BL/6 males were fed either control normal protein, normal carbohydrate diet (NP-NC; 18% casein, 21% sucrose, 42% corn starch, 10% corn oil; $n = 16$ entire and 8 vasectomised males) or isocaloric low protein, high carbohydrate diet (LP-HC; 9% casein, 24% sucrose, 49% corn starch, 10% corn oil; $n = 16$ entire and 8 vasectomised males) for a period of 8–12 weeks. Diets were manufactured commercially (Special Dietary Services Ltd; UK) and their composition described previously[2].

**F1 offspring generation.** Virgin 8-week-old female C57BL/6 mice ($n = 8$ litters per treatment) were super-ovulated by intraperitoneal injections of pregnant mare serum gonadotrophin (1 IU) and human chorionic gonadotrophin (1 IU) 46–48 h later. Intact NP-NC and LP-HC fed males were culled by cervical dislocation after a minimum of 8 weeks on respective diets. Sperm were isolated from caudal epididymi of NP-NC and LP-HC sires as described[2,15] and allowed to capacitate in vitro (37 °C, 135 mM NaCl, 5 mM KCl, 1 mM MgSO4, 2 mM CaCl2, 30 mM HEPES; supplemented immediately before use with 10 mM lactic acid, 1 mM sodium pyruvate, 20 mg/mL BSA, 25 mM NaHCO3). Females were artificially inseminated 12 h post human chorionic gonadotrophin injection with ~$10^7$ sperm and subsequently housed overnight with a vasectomized C57BL/6 male fed either NP-NC or LP-HC diet. Females were weighed regularly (every 4–5 days) for the detection of weight gain associated with a developing pregnancy. Four groups of offspring were generated, termed NN (NP-NC sperm and NP-NC seminal plasma), LL (LP-HC sperm and LP-HC seminal plasma), NL (NP-NC sperm and LP-HC seminal plasma) and LN (LP-HC sperm and NP-NC seminal plasma). The number of females inseminated, pregnancy rates, gestation lengths and litter parameters have been reported[2]. In the current study, we focused on tissues collected from F1 and F2 NN (NL-NC) and LL (LP-HC) groups as these provide a model for normal- and high carbohydrate intake in humans, and in order to reduce complicating factors.

**F2 offspring generation.** Sixteen-week-old F1 males ($n = 6$ males per treatment group; each from a different litter) were mated naturally to virgin, 8-week-old female C57BL/6 mice acquired separately for mating with F1 males. Females were allowed to develop to term and all dams and F2 offspring received standard chow and water ad libitum.

**Tissue collection.** F1 offspring were culled by cervical dislocation at either 3 (juvenile) or 16 (adult) weeks of age, whereas all F2 offspring were culled by cervical dislocation at 3 weeks of age. Blood samples were taken via heart puncture, centrifuged at $8k \times g$ (4 °C, 10 min) and the serum aliquoted and stored at −80 °C. Liver, brain, heart and adipose were dissected, weighed, snap frozen and stored at −80 °C.

**Stock solutions.**

1. GCTU. Guanidine (6 M guanidinium chloride) and thiourea (1.5 M) were dissolved in deionised $H_2O$ together and stored at room temperature out of direct sunlight.
2. DMT. Dichloromethane (3 parts), methanol (1 part) and triethylammonium chloride (0.002 parts, i.e. 500 mg/L) were mixed and stored at room temperature out of direct sunlight.
3. MS-mix. Propan-2-ol (2 parts) was mixed with methanol (1 part) and used to produce a solution of $CH_3COO.NH_4$ (7.5 mM).

**Tissue sample preparation and extraction of the lipid fraction.** Whole tissue/organ samples were prepared and extracted as described recently[22]. Solutions of homogenized organ preparations were injected into a well (96 well plate, Esslab Plate+™, 2.4 mL/well, glass-coated) followed by methanol spiked with internal standards (150 μL, internal standards shown in Table S5), water (500 μL) and DMT (500 μL) using a 96 channel pipette. The mixture was agitated (96 channel pipette) before being centrifuged ($3.2k \times g$, 2 min). A portion of the organic solution (20 μL) was transferred to a high throughput plate (384 well, glass-coated, Esslab Plate+™) before being dried ($N_{2(g)}$). When $4 \times 96$ well plates had been placed in the 384 well and the instrument was available, the dried films were re-dissolved (tert-

butylmethyl ether, 20 μL/well, and MS-mix, 80 μL/well) and the plate was heat-sealed and queued immediately, with the first injection within 10 min.

Samples with a high concentration of triglycerides (TGs; e.g. adipose tissue) were also treated to concentrate the phospholipid fraction so it too could be profiled[22]. A second portion of the organic phase from the extraction (100 μL) of was transferred to a shallow plate (96 well, glass-coated) before being dried ($N_{2 (g)}$), washed (hexane, $3 \times 100$ μL/well) and re-dissolved (DMT, 30 μL). The samples were transferred immediately to the high throughput analytical plate as above and dried ($N_{2(g)}$).

**Direct infusion mass spectrometry (DI-MS).** All samples were infused into an Exactive Orbitrap (Thermo, Hemel Hampstead, UK), using a TriVersa NanoMate (Advion, Ithaca US), for direct infusion mass spectrometry (DI-MS[21]). Samples (15 μL ea.) were sprayed at 1.2 kV in the positive ion mode. The Exactive started acquiring data 20 s after sample aspiration began. The Exactive acquired data with a scan rate of 1 Hz (resulting in a mass resolution of 100,000 full width at half-maximum [fwhm] at 400 $m/z$). The Automatic Gain Control was set to 3,000,000 and the maximum ion injection time to 50 ms. After 72 s of acquisition in positive mode the NanoMate and the Exactive switched over to negative ionization mode, decreasing the voltage to −1.5 kV and the maximum ion injection time to 50 ms. The spray was maintained for another 66 s, after which the NanoMate and Exactive switched over to negative mode with collision-induced dissociation (CID, 70 eV) for a further 66 s. After this time, the spray was stopped and the tip discarded, before the analysis of the next sample began. The sample plate was kept at 15 °C throughout the acquisition. Samples were run in row order. The instrument was operated in full scan mode from $m/z$ 150–1200 Da.

**DI-MS Data processing.** The lipid signals obtained were relative abundance ('semi-quantitative'), with the signal intensity of each lipid expressed relative to the total lipid signal intensity, for each individual, per mille (‰). Raw high-resolution mass-spectrometry data were processed using XCMS (www.bioconductor.org) and Peakpicker v 2.0 (an in-house R script[21]). Lists of known species (by $m/z$) were used for both positive ion and negative ionisation mode (~8k species). Signals that deviated by more than 9 ppm were discarded, as were those with a signal/noise ratio of <3 and those pertaining to fewer than 50% of samples. The correlation of signal intensity to concentration of the variable in QC samples (plasma, liver homogenate, brain homogenate, milk-formula mixture[42]; 0.25, 0.5, 1.0×) was used to identify which lipid signals were proportional to abundance in the sample type and volume used (threshold for acceptance was a correlation of >0.75). Signals were then signal corrected (divided by the sum of signals for that sample not including internal standards), in order to be able to compare samples in a manner unconfounded by total lipid mass. All statistical calculations were done on these finalised values. Annotations of the $m/z$ signals identified are listed in Supplementary Data 5. '(PW)' refers to adipose that was washed with petroleum spirit; the data from petrol-washed samples were used for negative ionisation mode (in which phospholipids are measured) where untreated samples were used for positive ionisation mode (in which triglycerides and their in-source fragmentation products, principally diglycerides, were measured). Lipid identification: 586 lipid variables in positive ionisation mode and up to 564 lipid variables in negative ionisation mode in liver, brain, heart and adipose homogenates and in serum were putatively identified according to the Metabolomics Standards Initiative at level 2.

**Lipid extraction and sample preparations for $^{31}P$ NMR.** The extraction of larger sample volumes for NMR was based on a method described previously[22,23]. Tissue homogenates were combined to give 5–10 mg of phospholipid per NMR sample. The samples of serum and prepared brain tissues from all groups were pooled and GCTU (250 μL) added to serum mixtures. Pooled solutions (5–8 mL) were diluted (DMT, 15 mL) and made uniphasic (methanol, 15 mL). The mixture was agitated and diluted and made biphasic (dichloromethane, 10 mL) before centrifugation ($3.2k \times g$, 2 min). The aqueous portion and any mesophasic solid was removed and discarded, and the organic solution dried under a flow of nitrogen. Samples were stored at −80 °C. Samples were dissolved in a modified[22] form of the 'CUBO' solvent system[43–46] (the amount of dueteriated dimethylformamide $d_7$-DMF was minimised). Stock solutions of the solvent consisted of dimethylformamide (3.5 mL), $d_7$-DMF (1.5 mL), triethylamine (1.5 mL) and guanidinium chloride (500 mg). Wilmad® 507PP tubes were used. Sample concentration was 5–10 mg lipids per sample (600 μL).

**NMR spectrometer and probe.** Lipid samples were run on a Bruker Avance Neo 800 MHz spectrometer, equipped with a QCI cryoprobe probe. 1D Phosphorous experiments were acquired using inverse gated proton decoupling. Spectra were averaged over 1312 transients with 3882 complex points with a spectral width of 14.98 ppm. An overall recovery delay of 8.4 s was used. Data were processed using an exponential line broadening window function of 1.5 Hz prior to zero filling to 32,768 points and Fourier transform. Data were processed and deconvoluted using TopSpin 4.0.7. Subsequent integrations above a noise threshold of 0.01% of the total $^{31}P$ were used to establish the relative molar quantity of a given phosphorus environment. A survey of $^{31}P$ traces is in Supplementary Data 2.

**Interpretation of profiling data and preparation of final lipidomics sheets.**
Dual spectroscopy[22] was used to interpret lipidomics data. Specifically: [31]P NMR data of hearts and livers from all generations and both phenotypes were collected and assigned (according to refs. [22,23,43–46]) and compared and found to be much more similar to one another than other sample types (tissues/compartments). Only a small number of representative, pooled samples from the CNS and serum were therefore run. One liver sample (F2N, NP-NC) was run twice, 48 h apart, to assess degradation within the sample. It was found that a small change in the abundance of *lyso*-PC was just measurable in this time, suggesting that sample preparation and running (<72 h) was sound. One large scale petrol-wash[22] was done on an adipose sample (F1A, LP-HC). These spectra were used to check for sample degradation in handling (e.g. appearance of PA) and inform assignments of signals measured using DI-MS. For example, [31]P NMR shows that serum has at least 50× PC more than PE, with very little or no PS, indicating that the balance of probabilities for assignments falls on the PC rather than the isobaric PE (positive ionisation mode) or PS (negative ionisation mode) isoform. These spectra were also used to interpret the difference in ionisation efficiency between species. These data show that the ionisation efficiency of *lyso*-PC and *lyso*-PE are both very high in negative ionisation mode, where that of sphingomyelin is under-represented in both ionisation modes.

**Statistics and reproducibility.** Univariate and bivariate statistical calculations were made using Microsoft Excel 2016, as were calculations of Eqs. 1 and 2. Graphs were prepared in OriginLab 2018 or Excel 2016 from mean (including Eq. 1) and standard deviation or error-normalised fold change (Eq. 2) as appropriate. Equations 1 and 2 were generated de novo in the present study. Jaccard-Tanimoto Coefficients (JTCs) were used as a non-parametric measure of the distinctions between lipid variables associated with phenotype(s)[24,25]. The associated $p$ values were calculated following Rahman et al.[47]. The $p$ value associated with each $J$ represents the probability that the difference between the lists of variables for the two phenotypes occurred by random chance and should not be confused with $p$ values from the Student's $t$-test. The $p$ values that are associated with the Student's $t$-tests (Abundance Analyses) were interpreted using a corrected $p$ value of 0.0021 based on 586 dependent variables[40] as only variables in positive ionisation mode were used. Only lipid variables with a $p$ value below this and that were relevant to the hypothesis were used.

**Lipid traffic analysis.** The tissues used were mapped to the known biological/metabolic network (Fig. 1b). Lipid variables in each compartment (lipid station) were categorised according to whether they are unique to it (*U* type lipids), shared with one adjacent to it (*B* type lipids, uni- and bidirectional) or found in all compartments (*A* type lipids), as shown in Fig. 2a. The full switch analysis is shown in Supplementary Data 1 and code in S3. Dimensions for the abundance analysis were calculated using Eqs. 1 and 2 (vide supra). Variables were regarded as present if they had a signal strength >0 in ≥50% of samples of either phenotype group. The full abundance analysis is shown in Supplementary Data 4.

Novel code for the binary traffic analysis (for the switch analysis) and multinary traffic analysis (abundance analysis) was written in R (v3.6.x) and processed in RStudio (v1.2.5x). The full code (Lipid Traffic Analysis v1.0) can be found in the Supplementary Information. Briefly, MS signals data in Excel-readable *.csv format was uploaded with removal of the metadata (organ, extraction, plate location, enumeration of mass/charge ratios [m/z]), giving n (rows of observations) versus p (columns of lipids) of signal data. Layered functions were used to identify which variables were present in all (*A*), adjacent (*B*) or single (*U*) compartments.

For each observation, the detection of the signal data commenced initially with FALSE representing no lipid signal (NA) and TRUE representing abundance of a lipid (above a signal threshold). For a particular compartment (tissue/pool/station), all observations were sampled into a single binary vector of presence and absences. The detection was performed using non-redundant lipid names. The function Reduce(intersect, list (…)) represented the common lipids for a given axis. Matched lipids were obtained across each pool to identify the common intersection, SetA(All). Positive and negative ionisation mode mass spectrometry data were run in blocks in series. The lists of lipids for the NP-NC and LP-HC groups were processed for the common intersection giving SetA (*A*, ubiquitous lipid variables), SetB (*B*, lipid variables found in two adjacent compartments) and SetU (*U*, unique, for lipid variables found in one compartment but not its neighbours).

**Reporting summary.** Further information on research design is available in the Nature Research Reporting Summary linked to this article.

## Data availability
The full switch analysis of all generation groups used in this study is show in Supplementary Data 1. The NMR data acquired in the study is included in Supplementary Data 2. The full abundance analysis is shown in Supplementary Data 4. The MS dataset generated in the present study is available from the corresponding author on reasonable request, with annotations of signals shown in Supplementary Data 5.

## Code availability
The novel R code developed in the present study is in Supplementary Data 3 and is available publicly available through Zenodo[48] and BioRxiv[49].

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

## Acknowledgements

The authors wish to thank Drs S.G. Snowden and A. Sleigh for helpful discussions. S.F. would like to dedicate this paper to cryptographer W. Gordon Welchman (1906–1985) whose work on Traffic Analysis for wartime communications inspired the method for characterising lipid traffic described here. The authors also gratefully acknowledge funding from the BBSRC (BB/M027252/1 and BB/M027252/2 for S.F.). A.J.W. was supported by an Aston Research Centre for a Healthy Ageing fellowship.

## Author contributions

A.J.W. developed the mouse model, did all animal work and produced all tissue samples. S.F. conceived and supervised the network analysis and hypothesis, did all lipidomics, analysed data and wrote the manuscript. J.S. and D.C. designed the code and developed the lipid categories. D.C. wrote the code with N.H. and J.S. H.E.L.W. collected and analysed N.M.R. data with S.F. S.F. and A.K. designed the experiments. A.K., J.S. and A.J.W. wrote the original grant proposals. S.F. and A.K. analysed and interpreted data and revised the manuscript with comments from all authors. All authors commented on the manuscript and approved the final version.

## Competing interests

The authors declare no competing interests.
