## [Peer Review File · Communications Biology]

Reviewers' Comments:

Reviewer #1:

Remarks to the Author:

The manuscript of Furse et al entitled "Novel network analysis reveals that a high carbohydrate intake by fathers modulates lipid metabolism two generations hence" describes an interesting and novel approach of lipidomics data analysis from a mouse trial, highlighting how their tool 'Lipid Traffic Analysis' can be used to interpret alterations in lipid metabolism. The approach looks to find differences in lipid composition and abundance in different tissues from an intergenerational dietary intervention animal study and link these together in a novel network framework/flow.

The findings are interesting, and the concept of the Traffic Analysis tool is a novel and an interesting idea which has highlighted changes in lipid metabolism that occur across pathways (TG/PL) and with components from different sources (endogenous/dietary). This approach is of interest to the wider community and this tool may help in the difficult interpretation of multi-zone lipidomics data. I support the publication of this manuscript, however, I feel there does need to be some considerable revision of the results and discussion for enhanced clarity of the results. For example, there is very little discussion on the biological meaning of why in fig 5b there are only 3 phospholipids mentioned and that 2 of these are odd chain.

The authors should put more depth into the discussion of these interesting findings - what does it mean? How can this be? Why biologically does this happen? The authors states "This suggested there were several modulations to PC and PE metabolism associated with this phenotype, and is consistent with long-standing evidence that PC is used as a means for storing/transporting polyunsaturated FAs such as arachidonic acid²⁶" but this is vague to the reader.

Likewise, for figure 3b, "There were about four variables that distinguished the NP-NC Serum-Heart axis and the LP-HC axis, all of which also distinguished the Serum-Brain axis in the LP-HC group from the NP-NC (Fig. 3B). " is vague. Further interpretation of the biological significance of this finding would be useful. Also, the term 'about four variables' is vague.

The statistical methods, mass spec methods, etc are all appropriate and of high quality, and well described so that someone could reproduce the work, although the inhouse lipid identification tool may be needed if not already published.

Reviewer #2:

Remarks to the Author:

This study aims to investigate the impact of high paternal carbohydrate intake on offspring's phospholipid and triglyceride metabolism in F1 and F2 generations evaluating the lipid profiles in serum, liver, brain, heart and abdominal adipose tissues. After the identification using Mass Spectrometry and Nuclear Magnetic Resonance, associated with these two phenotypes (NP-NC, control; LP-HC, experimental) were analyzed using Lipid Traffic Analysis tool for analyzing lipid metabolism as a network. This analyzes showed that the type and abundance of lipid variables in and between tissues (known collectively as lipid traffic) differed between phenotypes and generation, suggesting that triglyceride (TG) and phosphatidylcholine (PC) metabolism were altered throughout the network by the nutritional programming, and over two generations. The network approach to the analysis of lipid metabolism was essential for identifying changes in lipid metabolism that occur across pathways (TG/PL), however more consistent data, like genetic studies, are needed to prove correlation of carbohydrate-rich consumption with 2 generation metabolic programming. However, I encourage the authors to complement the study because this being a work well depth, particularly useful for metabolic studies platform as it allows the effect of phenotypes to be understood the lipids correlation in different organ, systems, and thus compared.

Major comments:

- More consistent data must be provided to strengthen the hypothesis that the impact of high paternal carbohydrate intake on offspring's phospholipid and triglyceride metabolism in F1 and F2 generations.

Minor comments:

- Review the figures legends and codes, as well as their citations in the text, such as Figure 1 and paragraph 64-70, that the descript text is not observed in figure 1.

- Classify the identified lipids according to the metabolomics standards initiative (Sumner et al., 2007)

Reviewer #3:

Remarks to the Author:

The authors present a manuscript entitled "Novel network analysis reveals that a high carbohydrate intake by fathers modulates lipid metabolism two generations hence" in which they employ a novel lipid computational tool called "Lipid traffic Analysis" to map the movement of lipids across peripheral and central tissues. To demonstrate the utility of said technique the authors go on to demonstrate (to some extent) that a grandsires diet can influence lipid metabolism in subsequent offspring in up to two future generations. The manuscript is generally well written and the results that are presented are well written, however I do have some concerns. I think the authors would have benefited from a much more comprehensive analytical tool such as the Lipidizer (Sciex) for such an experiment, with confident identifications. This would make the Lipid tracking software much more impactful in my opinion. Overall, it is an interesting tool and could have some really useful future applications if made available to the scientific community.

Major Concerns:

To address the elephant in the room, the sample size used herein is very small to extrapolate such conclusions. I would advise being much more conservative.

The authors are correct in how they describe their measured features ("lipid variables") but I am concerned that no identification has been completed here other than their knowledge that certain lipids are measured in +ve/-ve modes of acquisition. When referring to a specific lipid species how can they be sure that the lipids they measure fall within that specific group? For instance, the authors make note of several isoforms of PC's. How can they be certain that these are correct identifications? This would need to be addressed and compounds labelled and scored according to the Metabolomics Standards Initiative.

Minor Concerns:

Why were all tissue types not available for all the offspring? See the legend for figure 1. Surely it would have been more valuable to investigate both hemispheres and as in the F1N generation? This is confusing and the authors need to explain why only certain tissues were available for certain progeny.

Figure 3 tells us in the caption header that everything is measured in +ve mode, but on further reading into the legend, -ve mode is also used for phosphatidylcholine.

An "Exactive Orbitrap" or a Q-Exactive?

Response to reviewer comments for Furse et al. 'Novel network analysis reveals that a high carbohydrate intake by fathers modulates lipid metabolism two generations hence'

We are very grateful for the reviewer comments on this manuscript and have endeavoured to implement the appropriate modifications. A point-by-point response to comments is below, including details of modifications.

Reviewers' comments:

Reviewer #1 (Remarks to the Author):

The manuscript of Furse et al entitled "Novel network analysis reveals that a high carbohydrate intake by fathers modulates lipid metabolism two generations hence" describes an interesting and novel approach of lipidomics data analysis from a mouse trial, highlighting how their tool 'Lipid Traffic Analysis' can be used to interpret alterations in lipid metabolism. The approach looks to find differences in lipid composition and abundance in different tissues from an intergenerational dietary intervention animal study and link these together in a novel network framework/flow. The findings are interesting, and the concept of the Traffic Analysis tool is a novel and an interesting idea which has highlighted changes in lipid metabolism that occur across pathways (TG/PL) and with components from different sources (endogenous/dietary). This approach is of interest to the wider community and this tool may help in the difficult interpretation of multi-zone lipidomics data. I support the publication of this manuscript

We are delighted with these comments and are grateful for this view of our work.

However, I feel there does need to be some considerable revision of the results and discussion for enhanced clarity of the results. For example, there is very little discussion on the biological meaning of why in fig 5b there are only 3 phospholipids mentioned and that 2 of these are odd chain. The authors should put more depth into the discussion of these interesting findings - what does it mean? How can this be? Why biologically does this happen? The authors states "This suggested there were several modulations to PC and PE metabolism associated with this phenotype, and is consistent with long-standing evidence that PC is used as a means for storing/transporting polyunsaturated FAs such as arachidonic acid²⁶" but this is vague to the reader.

The reviewer makes an important point. We have thoroughly revised the results and discussion sections. We have provided further explanation of the purpose of Fig. 5B, which is to highlight how lipid traffic differs between the phenotypes and so we agree that discussion here in particular is useful. The explanation of the purpose of network connections (routing) diagrams such as that in Fig 5B is made in the text where the first one appears (Fig. 3B):

'We plotted the presence of the appropriate variables in the switch analysis as a network connections diagram similar to those used in electronics and transport (Fig. 3B). This type of plot shows the routing or gating of the variables and is thus referred to here as a routing diagram. This analysis showed that...'

We have added the following to the Results around Fig. 5B:

'A routing diagram showed that two of these were re-routed to the CNS in the LP-HC (e.g. PC(43:2), Fig. 5B). This analysis suggests that several effects differentiate the two groups through the system, including biosynthesis and transport. It also showed that the sections of the CNS tested did not show the same shifts in distribution according to phenotype. This shows that distribution within the CNS is also altered as well as in the periphery. For example, we found that...'

We preferred to be circumspect in our analysis of the meaning of the results in order not over-interpret, however we include the following paragraph discussing the meaning and possible further work of the highlighted differences:

'There are several changes in the lipid traffic of phospholipids that differ from those of triglyceride variables. We have found evidence for changes to the traffic of both PC and PE (*Fig. 5*). Changes to PC traffic centre on the CNS and heart, with some striking differences in the gating of some phospholipid variables between phenotypes. The examples shown in *Fig. 5B* show that the gating of polyunsaturated and odd-chain-containing phospholipids is altered between phenotypes. We suggest that this evidence shows that several factors differ between phenotypes, many of which are worthy of further investigation. First, biosynthesis in the liver is affected. This changes the availability of these lipids for trafficking to other parts of the system. Second, some lipids are produced in the liver but do not appear elsewhere, suggesting that transport of lipids from the liver into the circulation is altered between phenotypes. Third, the termini in which they are found (heart/CNS) differs between feeding groups. This suggests uptake in tissues differs between the two groups. Between them, these changes show that a number of aspects of lipid metabolism and distribution are altered as a result of paternal dietary intake. Further studies in this area could include an analysis of gene expression of proteins involved in these processes, such as transporters in the blood-brain barrier that are known to transport polyunsaturated-fatty-acid-containing PCs into the CNS (refs). This could also be used gain insight into the expression of lipases that shuffle fatty acids between phospholipids (refs) and into proteins that remove single carbons from fatty acids to give odd-chain-containing species such as PC(43:2), *e.g. Hacl1* (refs).'

We have also made a number of smaller changes throughout the manuscript, including in the results section, to clarify the salient results and facilitate an analysis of what this means for the metabolism of the system.

Likewise, for figure 3b, "There were about four variables that distinguished the NP-NC Serum-Heart axis and the LP-HC axis, all of which also distinguished the Serum-Brain axis in the LP-HC group from the NP-NC (*Fig. 3B*). " is vague. Further interpretation of the biological significance of this finding would be useful.

We agree that this figure contains important results and welcome the suggestion to discuss them further. We have widened discussion of the results concerning triglycerides (*Figs 3B* and *4A, B*). We have included a concise explanation of the routing/gating diagrams (*Figs 3B* and *5B*). We have expanded the first two sentences of the third paragraph of the Discussion. They now read:

'The change in traffic of TGs offers insight into evidence for changes to metabolic control as a result of TGs crossing the blood-brain barrier. The phenomenon of metabolic effects associated with TGs in the CNS has been observed through central leptin and insulin receptor resistance (ref), however the variety of TGs has not previously been described. The present study shows that TGs associated with dietary intake are routed to the CNS in F1Ns whose fathers ate a high carbohydrate diet (LP-HC), whereas they are routed to the heart in a normal diet (TG(52:2, 56:4, 56:5), *Fig. 3B*). This result is illuminating because it shows that it is not simply a case of which species are present in dietary intake that shape how metabolism is organised. These data suggest that that intake shapes how lipid distribution is organised. This in turn hints at changes in infrastructure that offer a mechanism for possible changes in fuel uptake at the two termini (brain, heart) between phenotypes.

The same analysis in the F2N group showed that there were 14 odd-chain-containing TGs in the CNS of the NP-NC but not LP-HC group (Fig. 4B, Table S3), showing that changes to lipid distribution are evident at least two generations hence. This is strong evidence that the apparent controls over lipid distribution are associated with parental and even grand-parental diet.'

Also, the term 'about four variables' is vague.

We agree that 'about four variables' sounds vague. The exact number of variables involved is unclear as it consists of both TGs and DGs and we note that the DGs detected are the result of in-source fragmentation of TGs. We have therefore changed 'about' to 'at least' and clarified the relationship between TGs and DGs in DI-MS Data processing section of the Methods section.

The statistical methods, mass spec methods, etc are all appropriate and of high quality, and well described so that someone could reproduce the work, although the inhouse lipid identification tool may be needed if not already published.

We are delighted with these comments. Lipid identification is achieved through Peakpicker v2.0 (referred to in the text/references) and data processing approaches (using the QC samples, as described) and thus we suggest is published.

Reviewer #2 (Remarks to the Author):

This study aims to investigate the impact of high paternal carbohydrate intake on offspring's phospholipid and triglyceride metabolism in F1 and F2 generations evaluating the lipid profiles in serum, liver, brain, heart and abdominal adipose tissues. After the identification using Mass Spectrometry and Nuclear Magnetic Resonance, associated with these two phenotypes (NP-NC, control; LP-HC, experimental) were analyzed using Lipid Traffic Analysis tool for analyzing lipid metabolism as a network. This analyzes showed that the type and abundance of lipid variables in and between tissues (known collectively as lipid traffic) differed between phenotypes and generation, suggesting that triglyceride (TG) and phosphatidylcholine (PC) metabolism were altered throughout the network by the nutritional programming, and over two generations. The network approach to the analysis of lipid metabolism was essential for identifying changes in lipid metabolism that occur across pathways (TG/PL),

We are delighted to read these comments and are grateful for this analysis of our work.

however more consistent data, like genetic studies, are needed to prove correlation of carbohydrate-rich consumption with 2 generation metabolic programming. However, I encourage the authors to complement the study because this being a work well depth, particularly useful for metabolic studies platform as it allows the effect of phenotypes to be understood the lipids correlation in different organ, systems, and thus compared.

Major comments:

- More consistent data must be provided to strengthen the hypothesis that the impact of high paternal carbohydrate intake on offspring's phospholipid and triglyceride metabolism in F1 and F2 generations.

We think that the reviewer raises a useful and interesting point. The mechanisms driving nutritional programming are still mostly elusive. It is expected that epigenetic processes are involved but the direct epigenetic regulation of lipid metabolism need to be elucidated. Our work is providing the foundation to get to the metabolic mechanisms. The next step will be to make the link with epigenetic processes, but we see this as outside the scope of this work. We accept the limitation of only measuring the lipids in this respect and we have added this to discussion. It occurred to us that further work on gene expression in particular in these systems was an exciting possibility. We decided to limit the present study to a hypothesis-driven examination of lipid traffic as LTA is of course new and thus a deeper explanation of the technique and its construction is required. Furthermore, a suitably thorough investigation of gene expression in three generational groups would expand the dataset considerably, even when focused by the results from the LTA, making the resulting manuscript very large. However, we envisage that future studies in which focused LTA is used will incorporate techniques targeted techniques such as RNA-seq.

Minor comments:

- Review the figures legends and codes, as well as their citations in the text, such as Figure 1 and paragraph 64-70, that the descript text is not observed in figure 1.

We are grateful for this comment and have corrected the Figure citations.

- Classify the identified lipids according to the metabolomics standards initiative (Sumner et al., 2007)

We agree that standard classification according to the MSI is relevant to this study and being understood. The Internal standards used in data collection are listed in *Supplementary Table 1*, which allow us to characterise compound classes and identification of lipids are at level 2 of section 2.9 in Sumner *et al.* (2007). We have added a file to the Supplementary Information that shows all the variables identified for the positive and negative modes, with up to three annotations for each.

Reviewer #3 (Remarks to the Author):

The authors present a manuscript entitled “Novel network analysis reveals that a high carbohydrate intake by fathers modulates lipid metabolism two generations hence” in which they employ a novel lipid computational tool called “Lipid traffic Analysis” to map the movement of lipids across peripheral and central tissues. To demonstrate the utility of said technique the authors go on to demonstrate (to some extent) that a grandsires diet can influence lipid metabolism in subsequent offspring in up to two future generations. The manuscript is generally well written and the results that are presented are well written, however I do have some concerns. I think the authors would have benefited from a much more comprehensive analytical tool such as the Lipidizer (Sciex) for such an experiment, with confident identifications. This would make the Lipid tracking software much more impactful in my opinion. Overall, it is an interesting tool and could have some really useful future applications if made available to the scientific community.

We are grateful for these comments and find this analysis of our work useful.

The analytical tool mentioned by the reviewer is a well-known one that has proven useful in published studies. We are pleased to say that LTA is compatible with data from such tools and we envisage that

future uses of LTA will use data from Lipidizer or any better systems that are currently under development.

Major Concerns:

To address the elephant in the room, the sample size used herein is very small to extrapolate such conclusions. I would advise being much more conservative.

We agree that sample size is an important consideration in 'omics studies. The present study uses up to $n = 8$ per group for the F1 generation groups and $n = 6$ for F2, as described in Fig. 1. The group sizes were determined on the basis of power calculation for the testing of the primary outcome, which was changes in body size and cardiovascular or metabolic parameters. The fact that we were able to observe significant effects on the primary outcome (10.1113/jp278270, 10.1073/pnas.1806333115) showed that there had to be differences in metabolism, which we studied in molecular detail using this approach. We are therefore confident that group sizes of the magnitude used in the present study provide sufficient statistical power for the analyses presented and appropriate corrections for testing are used.

Following this comment and the desire of reviewer 1 for greater depth of discussion of the results, we have made a number of changes in the Results and Discussion sections that we argue support reasonable analysis of the data, in the context of its limits.

The authors are correct in how they describe their measured features ("lipid variables") but I am concerned that no identification has been completed here other than their knowledge that certain lipids are measured in +ve/-ve modes of acquisition. When referring to a specific lipid species how can they be sure that the lipids they measure fall within that specific group? For instance, the authors make note of several isoforms of PC's. How can they be certain that these are correct identifications? This would need to be addressed and compounds labelled and scored according to the Metabolomics Standards Initiative.

We agree that thorough identification of metabolites is important in omics studies. We agree that Direct Infusion MS is limited in its scope. One limit of the technique is that isobaric species cannot be distinguished within a given ionisation mode. One response to this is to use an orthogonal technique, which we note it is relatively uncommon in lipidomics. (We described this approach in 10.1007/s00216-020-02511-0, using the term dual spectroscopy). The abundance of head groups is measured unambiguously using ^{31}P NMR. This allow us to distinguish readily between PC and PE, two lipids that are isobaric in positive ionisation mode.

The reviewer mentions +ve and -ve ionisation modes of acquisition; we are careful to make the best of these. In the Switch Analysis, we used +ve mode for TGs as they are not generally isobaric for any other lipid classes and thus this mode is ideal. Negative ionisation mode was used for PCs, which are not isobaric with PEs, unlike in +ve mode. In our opinion, the phenomenon of isobars is a primary motivator for the use of LC-MS in lipidomics studies and thus this technique offers a means for overcoming this limit. However, we note firstly that this technique is still not able to give unambiguous structural information. Furthermore, this study comprised nearly 1200 samples across eight different tissue types. Our laboratory has begun to investigate the practicalities of higher throughput LCMS, as have others, however there are a number of challenges. For example, DIMS can be done through a 2 or 3 minute method and thus is it feasible to do 1200 samples in one analytical batch. 1200 samples run by LCMS and an 11 minute method cannot realistically be done in one analytical batch yet. However, we

anticipate that future studies will use LCMS as standard and we look forward to that innovation becoming standard. For the present, we have made use of both ionisation modes in DIMS in the present study and combined this with ³¹P NMR in order to give a reliable lipid profile of the samples used. The file we have added to the Supplementary Information that shows all the variables identified for the positive and negative modes (up to three annotations) may help assist readers in understanding the profile described.

Minor Concerns:

Why were all tissue types not available for all the offspring? See the legend for figure 1. Surely it would have been more valuable to investigate both hemispheres and as in the F1N generation? This is confusing and the authors need to explain why only certain tissues were available for certain progeny.

We recognise the difference in the tissues available for the different generational groups. This is shaped by a number of factors. First, the tissues used were those available after many other experiments had taken place. Four papers have been published, with two more under construction, that are based on data from experiments on tissues from the animals used. (10.1113/jp278270, 10.1073/pnas.1806333115, 10.1152/ajpheart.00981.2013, 10.1016/j.bbadis.2017.02.009). This includes, for example, separation of hemispheres of brains for histology. This limited the tissues that were available for the present study. Second, neonates typically have very little adipose tissue and thus this tissue was not dissectible and thus this compartment was not available for the present study (to our lasting regret!). Third, prompt and consistent dissection and freezing of tissues was a key priority. However, we agree that it would be useful to the reader for this aspect of our study to be clear. We previously stated in the caption of Fig. 1 'Adipose was only available for F1A groups, whole brain samples used for F1N groups, with separate right brain and cerebellum for F1A and F2N.' in order to make this clearer we have inserted the following text earlier in the caption, replacing it:

'Adipose tissue was not available for neonates and thus networks for F1N and F2N do not include this tissue. Cerebellum and right hemispheres of the brains of the F1A and F2N groups, enabling separate analysis of cortices and the cerebellum in these groups.'

We have also modified the text in the introduction to make this clear:

'...was used to analyse lipidomics data from liver, serum, brain, heart and adipose tissues for the F1A group (Fig. 1B). Liver, serum, brain and heart samples were used in neonate networks as adipose tissue was too small to be dissectible in these individuals.

Figure 3 tells us in the caption header that everything is measured in +ve mode, but on further reading into the legend, -ve mode is also used for phosphatidylcholine.

We are grateful for this being spotted, the original figure had only one panel but after adding the second we had failed to delete the appropriate part of the caption header. This has now been corrected.

An "Exactive Orbitrap" or a Q-Exactive?

We used and Exactive Orbitrap, not a Q-Exactive.

REVIEWERS' COMMENTS:

Reviewer #1 (Remarks to the Author):

The authors have made an excellent and substantial revision of this manuscript, and have done a good job of addressing the reviewer comments from the first review. I am happy with the extra information and examples around further interpretation and discussion of the results, and feel now that this is ready for publication.

The quality of the writing is high, the statistics are appropriate, and the authors are well aware of alternative techniques to do this work, and the limitations that their MS data have (thus including NMR as part of the suite of tools). This Traffic Analysis tool is going to be potentially very useful to the community, and it is applicable with other lipidomics approaches (actually, potentially even more valuable with the quantitative LCMS targeted approaches being used now by Peter Meikle and co). The authors have also noted and discussed the limitations of the sample number for a metabolomics experiment, however, the results are interpreted/claimed to an appropriate level, and the tool and novel approach is too valuable to not be published.

Reviewer #2 (Remarks to the Author):

The manuscript entitled "Novel network analysis reveals that a high carbohydrate intake by fathers modulates lipid metabolism two generations hence" of Furse et al describes an investigation of the impact of high paternal carbohydrate intake on offspring phospholipid and triglyceride metabolism in F1 and F2 generations, by analysis of serum, liver, brain, heart and abdominal adipose tissues using mouse model. The computational tool Lipid Traffic Analysis, was used to analyze the lipid 'highways' in the organism and identify altered lipid metabolism across organs.

The studies using Traffic Analysis tool to analyze lipid traffic after higher carbohydrate intake metabolism and the effects of metabolic disease across generations is very interesting, and this approach can help projects that work with combinations of lipids data sets.

However, some considerations still need to be corrected or added to the manuscript.

Major comments:

-As the epigenetic studies was not realized to prove the modulation through generation F1 e F2 should be considered change the Title of the manuscript to corroborate the idea that the correlation results between generations is a suggestion, and has not been effectively proven.

-The author have to be clearer how the lipid identification was realized.

-Add in the manuscript the level of identified lipids according to the metabolomics standards initiative (Sumner et al., 2007). (e.g. "...and 586 lipid variables were putatively identified according to the Metabolomics Standards Initiative as level 2...")

-In general, do not put an explanation in the figure legends, but in the text of the manuscript.

-Based on sample size and the lack of more in-depth studies that prove the effects of higher carbohydrate across generations, it is not possible to make this type of conclusion. "In conclusion, this study has shown that the hypothesis that lipid metabolism is altered in offspring as a result of unbalanced dietary intake". The authors need to make it clear that the data are indicative of this correlation and that further studies need to be carried out to be proven.

Minor comments:

-Panels A and B were not identified in Figure 1.

-Put in the legend the correlation of the colors with the control and the variable represented in the figures.

Reviewer #3 (Remarks to the Author):

The authors have addressed all of my concerns satisfactorily.

REVIEWERS' COMMENTS:

Reviewer #1 (Remarks to the Author):

The authors have made an excellent and substantial revision of this manuscript, and have done a good job of addressing the reviewer comments from the first review. I am happy with the extra information and examples around further interpretation and discussion of the results, and feel now that this is ready for publication. The quality of the writing is high, the statistics are appropriate, and the authors are well aware of alternative techniques to do this work, and the limitations that their MS data have (thus including NMR as part of the suite of tools). This Traffic Analysis tool is going to be potentially very useful to the community, and it is applicable with other lipidomics approaches (actually, potentially even more valuable with the quantitative LCMS targeted approaches being used now by Peter Meikle and co). The authors have also noted and discussed the limitations of the sample number for a metabolomics experiment, however, the results are interpreted/claimed to an appropriate level, and the tool and novel approach is too valuable to not be published.

We are delighted with these comments and are grateful for this feedback on our work.

Reviewer #2 (Remarks to the Author):

The manuscript entitled "Novel network analysis reveals that a high carbohydrate intake by fathers modulates lipid metabolism two generations hence" of Furse et al describes an investigation of the impact of high paternal carbohydrate intake on offspring phospholipid and triglyceride metabolism in F1 and F2 generations, by analysis of serum, liver, brain, heart and abdominal adipose tissues using mouse model. The computational tool Lipid Traffic Analysis, was used to analyze the lipid 'highways' in the organism and identify altered lipid metabolism across organs.

The studies using Traffic Analysis tool to analyze lipid traffic after higher carbohydrate intake metabolism and the effects of metabolic disease across generations is very interesting, and this approach can help projects that work with combinations of lipids data sets.

We are grateful for these comments and are pleased to receive feedback on our work. We have endeavoured to make the changes as requested below.

However, some considerations still need to be corrected or added to the manuscript.

Major comments:

-As the epigenetic studies was not realized to prove the modulation through generation F1 e F2 should be considered change the Title of the manuscript to corroborate the idea that the correlation results between generations is a suggestion, and has not been effectively proven.

The editorial team suggested changing the title to 'Lipid traffic analysis reveals the impact of high paternal carbohydrate intake on offsprings' lipid metabolism', something with which we agree and also think would discharge this request.

-The author have to be clearer how the lipid identification was realized.

We agree that lipid identification is an important stage in any analysis of lipid metabolism. Between the Methods and Results sections, we detail the approach used (dual spectroscopy) and its literature basis (DOI: 10.1007/s00216-020-02511-0). The process of assigning annotations to both 31P NMR and HR-MS signals is explained in the Methods section, showing how we used QC samples to identify reliable signals. Possible isobars have been listed in the supplementary data. We have also included a note on to what level in the metabolics standards initiative (see below).

-Add in the manuscript the level of identified lipids according to the metabolomics standards initiative (Sumner et al., 2007). (e.g. "...and 586 lipid variables were putatively identified according to the Metabolomics Standards Initiative as level 2...")

We agree that the level to which lipids are identified in the MSI is useful to readers and so have stated this explicitly in the DI-MS data processing section:

'Lipid identification: 586 lipid variables in positive ionisation mode and up to 564 lipid variables in negative ionisation mode in liver, brain, heart and adipose homogenates and in serum were putatively identified according to the Metabolomics Standards Initiative at level 2.'

-In general, do not put an explanation in the figure legends, but in the text of the manuscript.

We agree that careful preparation of figures is important. We have been careful to follow the journal's requirements on figure captions, paying special attention to the need for figures to be understandable without reference to the main text. We are also acutely aware that the Jaccard-Tanimoto distances used are not as common as Student's *t*-tests so have been careful to make these clear where they are used.

-Based on sample size and the lack of more in-depth studies that prove the effects of higher carbohydrate across generations, it is not possible to make this type of conclusion. "In conclusion, this study has shown that the hypothesis that lipid metabolism is altered in offspring as a result of unbalanced dietary intake". The authors need to make it clear that the data are indicative of this correlation and that further studies need to be carried out to be proven.

We agree that the way studies present their findings is important. We also note reviewer 1's comments that we have 'interpreted/claimed the results to an appropriate level'. We have therefore been careful in modifying the last sentences of the conclusion to satisfy both viewpoints. We use the last two sentences of the Conclusion to make clear that further work is required to take further action than identifying the lipids, lipid pathways and lipid distribution associated with the phenotypes:

'...however further work using transcriptomics techniques is required to understand changes to the infrastructure associated with lipid metabolism. These and other techniques can be used to show how the damaging changes to lipid metabolism that have been identified can be reversed.'

Minor comments:

-Panels A and B were not identified in Figure 1.

We are grateful for this comment and have inserted labels for the figure panels in this and all figures, according to journal formatting requirements.

-Put in the legend the correlation of the colors with the control and the variable represented in the figures.

We agree that clear figures are important in any manuscript. We have made reference to the use of blue and orange in each figure by using explanations such as 'Orange is used for the LP-HC group whereas blue is used for the NP-NC group.' As well as the existing visual renderings (inserted Key). There is therefore reference to the colour scheme used in each figure caption.

Reviewer #3 (Remarks to the Author):

The authors have addressed all of my concerns satisfactorily.

We are delighted that our revisions have been received satisfactorily.